# Piperlongumine Inhibits Titanium Particles-Induced Osteolysis, Osteoclast Formation, and RANKL-Induced Signaling Pathways

**DOI:** 10.3390/ijms23052868

**Published:** 2022-03-05

**Authors:** Xuan Liu, Li Diao, Yudie Zhang, Xue Yang, Junnan Zhou, Yuhang Mao, Xiaotian Shi, Fuli Zhao, Mei Liu

**Affiliations:** Jiangsu Key Laboratory for Molecular and Medical Biotechnology, College of Life Sciences, Nanjing Normal University, Nanjing 210023, China; liuxuan9751@163.com (X.L.); diaoli1125@163.com (L.D.); zyd9909119@163.com (Y.Z.); yangxue9264983129@163.com (X.Y.); zhoujunnannnu@163.com (J.Z.); sky-maoyuhang@nnu.edu.cn (Y.M.); shixiaotian2022@163.com (X.S.); zfl18761689227@163.com (F.Z.)

**Keywords:** piperlongumine, osteoclast formation, bone resorption, RANKL-induced signaling pathways, osteolysis

## Abstract

Wear particle-induced aseptic loosening is the most common complication of total joint arthroplasty (TJA). Excessive osteoclast formation and bone resorptive activation have been considered to be responsible for extensive bone destruction and prosthesis failure. Therefore, identification of anti-osteoclastogenesis agents is a potential therapy strategy for the treatment of aseptic loosening and other osteoclast-related osteolysis diseases. In the present study, we reported, for the first time, that piperlongumine (PL), a key alkaloid compound from *Piper longum* fruits, could significantly suppress the formation and activation of osteoclasts. Furthermore, PL effectively decreased the mRNA expressions of osteoclastic marker genes such as *tartrate-resistant acid phosphatase* (*TRAP*), *calcitonin receptor* (*CTR*), and *cathepsin K* (*CTSK*). In addition, PL suppressed the receptor activator of nuclear factor-κB ligand (RANKL)-induced activations of MAPKs (ERK, JNK and p38) and NF-κB, which down-regulated the protein expression of nuclear factor of activated T-cells cytoplasmic 1 (NFATc1). Using a titanium (Ti) particle-induced calvarial osteolysis model, we demonstrated that PL could ameliorate Ti particle-induced bone loss in vivo. These data provide strong evidence that PL has the potential to treat osteoclast-related diseases including periprosthetic osteolysis (PPO) and aseptic loosening.

## 1. Introduction

Total joint replacement (TJR) is an effective clinical treatment for treating end-stage joint diseases such as osteoarthritis (OA), rheumatoid arthritis (RA) and femoral head necrosis [1]. With the number of patients receiving this surgery constantly increasing, particle-induced periprosthetic osteolysis (PPO) and subsequent aseptic loosening have attracted more and more attention [2]. Recently, much research has focused on improving the implant designs [3], perfecting surgical techniques, and searching for better biomaterials. However, these strategies are unlikely to completely eliminate particle generation from implants [4]. These wear particles can induce excessive osteoclast formation and bone loss which is demonstrated to be one of the main reasons of aseptic loosening [5]. A growing body of evidence suggests that anti-osteoclastogenesis is a potential therapy strategy [6]. Many drugs such as Denosumab and Zoledronic Acid have been developed to inhibit osteoclastogenesis and, thereby, treat particle-induced peri-implant osteolysis [7]. However, these clinical drugs have been found to bring a series of side effects to patients after long-term use, including digestive system disorders, osteonecrosis, and even some types of malignancy [8,9,10].

As the well-known “king of spices”, *Piper longum* L. is widely used as a seasoning ingredient to enhance food flavor as well as in food preserving (Figure 1A,B). In addition to its culinary uses, *Piper longum* L. and its bioactive phyto-compounds have also been extensively explored for their pharmacological effects [11] (Figure 1C). Piperlongumine (PL) (Figure 1D), a key alkaloid compound from *Piper longum* fruits, has been documented to have subsets of biological properties, including antiplatelet aggregation, anti-inflammation, neuroprotection, anti-hyperlipidemia, and anti-tumor effect [12,13,14,15,16,17,18,19,20,21,22]. Notably, Sun et al. and Xiao et al. verified that PL could attenuate collagen-induced arthritis (CIA) through relieving joint inflammation and inhibiting bone damage [23,24]. It is well known that osteoclasts are specialized and sole bone-resorbing cells and play an indispensable role in bone destruction of RA. Thus, we hypothesized that PL may play a therapeutic role in osteoclast-mediated bone lytic diseases. In this study, we investigated the potential protection of PL against Ti particle-induced osteolysis in vivo and subsequently revealed the underlying molecular mechanisms of PL in osteoclastogenesis.

## 2. Results

### 2.1. PL Treatment Suppressed RANKL-Induced Osteoclastogenesis In Vitro

Since bone marrow-derived macrophage (BMM) cells were osteoclast progenitors, we, firstly, used these cells to examine the possible cytotoxic effect of PL by an MTX assay. The results showed that concentration of PL at and below 1 μM did not affect the viability of BMM cells, and the calculated IC_50_ for this effect was 6.35 μM (Figure 2A). In the subsequent in vitro experiments, concentrations of PL did not exceed 1 μM.

To investigate the effect of PL on RANKL-induced osteoclast formation, BMMs were treated with different doses of PL in the presence of RANKL. As shown in Figure 2B,C, PL significantly inhibited osteoclastogenesis in a dose-dependent manner in vitro, as evidenced by the reduced number and average area of TRAP^+^ multinucleated (≥3 nuclei) cells. To explore at which stage of osteoclastogenesis PL exerts its suppressive effect, we incubated BMM cells with 1 µM PL at different differentiation time points, as previously described [25]. The results showed that PL treatment on day three of the five-day culture (late treatment) and treatment on the first two days of RANKL induction (early treatment) had similar and significant inhibitory effect on osteoclast formation, both of which resulted in an approximate 30% reduction in formation of TRAP^+^ osteoclasts (Figure 2D,E). Notably, consecutive treatment with PL (early and late treatment) had a stronger suppression in the number and average area of osteoclasts (Figure 2D,E). To examine the possible effect of PL on apoptosis, a flow cytometry-based assay was performed. As shown in Figure 2F, the given doses of PL did not affect apoptosis, suggesting that the inhibitory effect of PL on osteoclast formation was not due to its apoptosis-inducing action. 

Since bone homeostasis is maintained by the balance between osteoclastic bone resorption and osteoblastic bone formation, we next used an in vitro osteoblastogenesis assay to investigate the effect of PL on osteoblast differentiation. The result showed that concentrations of PL that inhibited osteoclast formation (0.25~1 µM) had little effect on the expression of ALP (Figure 2G). Thus, in the subsequent studies, PL was chosen as a prototype to further examine its mechanism of action in osteoclastogenesis and therapeutic potential in vivo.

### 2.2. PL Inhibited the Formation of F-Actin Ring and Osteoclastic Resorption Function 

Since PL significantly inhibited RANKL-induced osteoclastogenesis, we then investigated whether PL could downregulate bone resorption function of osteoclasts. The scanning electron microscopy results showed that resorption trenches and resorption pits were both obviously observed in the control group. In addition, the extent of trenches reached approximately 60% of the resorbed surface. However, these bone resorption areas, especially the areas of trenches, were significantly decreased by PL treatment (Figure 3A,B).

It is well known that F-actin ring is an essential prerequisite for bone resorption [26]. Thus, we employed immunofluorescence analysis to further evaluate the effect of PL on formation of F-actin rings. As shown in Figure 3C,D, intact-structured F-actin rings were clearly observed in the control group. However, the numbers of F-actin rings and actin ring surface/cell surface were significantly decreased by PL treatment, which further confirmed that PL could inhibit the bone resportive function of osteoclasts.

### 2.3. PL Treatment Inhibited Osteoclastic-Related Genes Expression

To further confirm the inhibitory effect of PL on osteoclast formation and function, the mRNA levels of osteoclast-associated genes including *TRAP*, *CTR*, *CTSK*, *NFATc1*, and *c-Fos* were determined by Q-PCR. The results showed RANKL stimulation markedly increased the transcriptional levels of those above-mentioned genes (Figure 4). However, these increases were significantly inhibited by PL treatment (Figure 4), which further confirmed the suppressive effect of PL on RANKL-induced osteoclastogenesis.

### 2.4. PL Suppressed RANKL-Stimulated Activations of MAPK and NF-kB Signaling Pathways

In order to discover the molecular mechanisms of inhibition by PL on osteoclast formation and function, RANKL-stimulated MAPK and NF-κB pathways were tested by Western blot. As expected, RANKL stimulation dramatically increased the phosphorylation levels of the three MAPK family members, namely extracellular signal-regulated kinase (ERK), c-Jun N-terminal kinase (JNK), and p38 (Figure 5A,B). Notably, all these increases were significantly suppressed by 1 μM PL (Figure 5A,B). In addition, PL treatment significantly inhibited RANKL-stimulated degradation of inhibitor of NF-κB (IκBα) (Figure 5A,B), suggesting it also affected the NF-κB pathway.

NFATc1 is the master downstream transcription factor of MAPKs and NF-κB pathways for osteoclast formation and activity. Thus, we detected whether PL could affect the induction of NFATc1 following RANKL stimulation. As expected, the protein expression of NFATc1 was enhanced at day three after RANKL stimulation and was more pronounced at day five (Figure 5C,D). However, this enhancement was significantly inhibited by PL treatment (Figure 5C,D), which further confirmed the inhibitory action of PL on RANKL-induced activation of MAPK and NF-κB pathways.

### 2.5. PL Protected against Ti Particle-Induced Calvarial Osteolysis

The in vitro experiments have demonstrated that PL exerts an inhibitory effect on osteoclast formation and function, so, next, we explored the potential protection of PL against Ti particle-induced bone loss in mice calvarias. Three-dimensional reconstruction result showed that extensive surface erosions were induced by Ti on the clavaria of the vehicle-treated group. However, the severity of Ti particle-induced osteolysis was significantly inhibited by PL treatment (Figure 6A). Moreover, quantification of bone parameters confirmed that PL treatment significantly elevated BMD and BV/TV, both of which are important indicators directly reflecting bone mass (Figure 6B,C). To further confirm the therapeutic effect of PL on Ti particle-induced osteolysis, histological assessment and histomorphometric analysis were performed. As shown in Figure 6D, in contrast to the abundance of TRAP^+^ osteoclasts and associated extensive bone resorption in the vehicle-treated group, PL treatment significantly decreased the number of TRAP^+^ multinucleated cells with concomitant decrease in bone osteolysis. Moreover, the percentage of osteoclast surface relative to the bone surface (OcS/BS) was also reduced by PL treatment, which further demonstrated the protection of PL against Ti particle-induced calvarial osteolysis (Figure 6E,F).

## 3. Discussion

Particulate wear debris from implants can spread in the joint cavity and surface, and induce intricate biological responses including PPO and aseptic loosening [27]. As we know, wear particle can stimulate macrophages, T lymphocytes, fibroblasts, and foreign body giant cells to secrete TNFα, IL-1β, IL-6, prostaglandin E2, and other proinflammatory cytokines. These proinflammatory cytokines subsequently induce RANKL synthesis by several cells types such as osteoblasts, T cells, and marrow stromal cells. The increased level of RANKL at the prosthesis site enhances osteoclast formation and activation, thus shifting the local bone metabolic balance to over-activated bone loss. Therefore, inhibition of osteoclastogensis may serve as a therapeutic strategy for the treatment of PPO and aseptic loosening after TJA.

PL, a natural product isolated from *Piper longum* fruits, exhibits several biological activities such as anti-inflammatory and anti-tumor effects [14,18]. To the best of our knowledge, the potential effect of PL on osteoclast-related diseases has not yet been investigated. In the present study, we explored the effect of PL on osteoclast formation and function, and on a Ti particle-induced calvarial osteolysis model in vivo.

Our results showed that PL effectively inhibited RANKL-induced osteoclastogenesis, as evidenced by the reduction in the number and area of osteoclasts. In addition, immunofluorescence and SEM analysis both verified that PL significantly down-regulated the bone-resorpting function of osteoclasts. The data of MTS assay and flow cytometry further suggested that these suppressive actions of PL were not due to its effect on viability or apoptosis of BMMs. In concordance with the in vitro data, our in vivo experimental results showed that PL could significantly suppress Ti particle-induced bone loss. Micro-CT and histomorphological analysis demonstrated that bone erosion and inflammatory cell invasion were both alleviated in the PL-treated groups, especially in the PL high-dose group. These data provide strong evidence that PL has the potential to treat osteoclast-related diseases including PPO and aseptic loosening.

Having confirmed that PL plays inhibitory roles in RANKL-induced osteoclastogenesis and in Ti particle-induced osteolysis model, we further investigated the underlying molecular mechanisms. As we know, the molecular events invoked during early phase of osteoclastogenesis and late phase are different. In the early stage, RANK signaling is mediated by recruiting adaptor molecule such as TRAF6, which leads to the activation of MAPKs and NF-κB. Activated MAPKs and NF-κB induce NFATc1, which is the key osteoclastogenesis regulator [28]. In the late stage of osteoclastogenesis, the co-stimulatory signal induces Ca^2+^ oscillation via PLCγ2 together with c-Fos/AP-1, wherein Ca^2+^ signaling facilitates the robust production of NFATc1 [29]. Subsequently, NFATc1 translocates into the nucleus, where it induces numerous osteoclast-specific target genes that are responsible for cell fusion and function [30]. In this present study, we found that both early phage and late phage of osteoclastogenesis were inhibited by PL. As reported in similar research works [31,32,33,34], we mainly revealed the compound’s molecular mechanism of action on early differentiation of osteoclasts. We, firstly, explored the effect of PL on RANKL-induced MAPK pathways [35]. MAPK family members (ERK, JNK and p38) have been demonstrated to play important roles in osteoclast survival, differentiation and activation. Phosphorylation of JNK modulates the transcriptional activity of AP-1, which is composed of c-Jun and c-Fos, the latter being essential for osteoclast differentiation [36]. p38 MAPK-mediated signals have been reported to be essential for RANKL-induced osteoclast differentiation while ERK activation is important in IL-1 and TNFa-mediated osteoclast survival [37,38]. Phosphorylated ERK can translocate into nucleus to activate NFAT proteins, thereby inducing osteoclast differentiation and function [37,39]. In our study, PL could significantly suppress the phosphorylation levels of RANKL-induced JNK, ERK, and p38. The capability to inhibit all three MAPK pathways indicates that PL may target a common activating kinase or deactivating phosphatase. The mode of PL’s action on the degree of MAPK phosphorylation warrants further investigations. In addition to MAPK pathways, we also examined the effect of PL on the NF-κB pathway, which has been demonstrated to play a pivotal role in osteoclastogenesis and bone resorption [40]. Our result showed that PL significantly suppressed the degradation of IκBα, indicating that the inhibition of PL on osteoclastogenesis might attribute, at least in part, to the NF-κB pathway. All these molecular mechanism data suggest that PL may have multiple targets, and further investigations are needed to unveil its direct binding sites. 

Although PL’s inhibition on late phage of osteoclastogenesis has not been fully explored, we did examine the effect of PL on the induction of NFATc1, which has been considered to represent a master switch for regulating late differentiation of osteoclasts. Additionally, we also detected the transcripts of NFATc1’s downstream target genes such as *TRAP*, *CTR* and *CTSK*, all of which are late differentiation marker genes. As expected, PL effectively suppressed the induction of NFATc1 and decreased the transcripts of osteoclastic marker genes. Notably, NFATc1 also can be spatio-temporally induced by RANKL-stimulated MAPK and NF-κB signaling pathways [41], therefore, further investigations are needed to unveil the molecular effect of PL on osteoclast differentiation, especially on the late phage of osteoclastogenesis. 

In conclusion, our study showed that PL could inhibit RANKL-induced osteoclast formation and function via NFATc1 by blocking MAPKs (ERK, JNK and p38) and NF-κB signaling pathways. Furthermore, PL attenuated Ti particle-induced osteolysis in vivo. These data suggest that PL has the potential to be developed as a treatment for aseptic loosening and other osteoclast-related osteolytic diseases.

## 4. Materials and Methods

### 4.1. Media and Reagents

PL (C_17_H_19_NO_5_, Purity ≥ 99.19%) (Figure 1D) was obtained from MedchemExpress (Shanghai, China). M-CSF and recombinant mouse RANKL were purchased from R&D Systems (Minneapolis, MN, USA). Eagle’s minimum essential medium, alpha modification (α-MEM), and penicillin/streptomycin were purchased from Gibco (Grand Island, NE, USA). Fetal bovine serum (FBS) was obtained from Invigentech Inc. (Irvine, CA, USA). Annexin V-FITC/propidium iodide (PI) apoptosis detection kit was from KenGEN Biotech. Co., Ltd. (Nanjing, China). Dexamethasone, β-glycerophosphate, ascorbic acid, MTS reagents, and TRAP staining kit were obtained from Sigma-Aldrich (St. Louis, MO, USA). Alkaline phosphatase (ALP) staining kit was obtained from Beyotime Biotechnology Inc. (Shanghai, China). TRIzol reagent was purchased from Vazyme Biotech Co., Ltd. (Nanjing, China). Primary antibodies targeting p-ERK, p-JNK, p-p38, total ERK, total JNK, total p38, IκBα, and GAPDH were from Cell Signaling Technology, Inc. (Beverly, MA, USA). NFATc1 antibody was purchased from BD Biosciences (San Jose, CA, USA). Ti particles (1–3 mm in diameter) were purchased from Johnson Matthey Company (Ward Hill, MA, USA). In order to remove possible adherent endotoxin, these particles were washed with 70% ethanol for 48 h and then heated at 250 °C for 3 h. Subsequently, Ti particles were resuspended in sterile PBS at a concentration of 300 mg/mL and stored at 4 °C until for use.

### 4.2. Cell Culture and Osteoclast Differentiation

Mouse BMM cells were isolated, purified, and cultured, per our previous report [4]. In brief, 6–8-week-old male C57BL/6 mice were killed by cervical dislocation. Cells from tibiae and femurs were collected and incubated with α-MEM complete medium (α-MEM with 10% FBS and 1% penicillin/streptomycin) containing 30 ng/mL M-CSF at 37 °C. After a 5-day culture, the adherent cells were used as BMM cells. BMM cells were plated into 96-well plates (8 × 10^3^ per well) and treated with varying doses of PL (0, 0.25, 0.5 and 1 µM) in the presence of M-CSF (30 ng/mL) and RANKL (100 ng/mL) for 5 days. The cells were fixed with 4% paraformaldehyde and stained for TRAP activity. The numbers and areas of TRAP^+^ multinucleated cells (nuclei ≥ 3) were determined using ImageJ software.

### 4.3. Cell Viability Assay

The cytotoxic effect of PL on BMM cells was assessed by an MTS assay. In brief, BMM cells (1 × 10^4^ cells/well) were seeded into 96-well plates and cultured overnight with α-MEM complete medium containing 30 ng/mL M-CSF. The cells were incubated with different concentrations of PL (0, 0.5, 1, 2, 4, 8, 16, and 32 μM) for 48 h and MTS/PMS mixture was added to each well for another 4 h. The absorbance at 490 nm was measured using a microplate reader (BMG LABTECH GmbH, Ortenberg, Germany), and the half-maximal inhibitory concentration (IC_50_) was counted.

### 4.4. Apoptosis Assay

The apoptosis-inducing effect of PL on BMM cells was investigated using an annexin V-FITC/PI apoptosis detection kit. In accordance with the instructions of the manufacturer, BMM cells (1 × 10^6^ cells/well) were seeded into 6-well plates and treated with different doses of PL (0, 0.25, 0.5, and 1 µM) for 24 h. Cells were collected by centrifugation and stained with annexin V and PI solution. The apoptotic cells were detected by a flow cytometer (Beckman Coulter, CA, USA) and the apoptotic rates were analyzed using Cell Quest Software (FACScan; Becton Dickinson, Franklin Lakes, NJ, USA).

### 4.5. In Vitro Osteoblast Differentiation

Human osteoblasts (hFOB 1.19) were purchased from the Cell Bank of the Chinese Academy of Sciences (Shanghai, China). These cells were cultured in DMEM/F12 complete culture medium in a humidified atmosphere of 5% CO_2_. In vitro osteoblastogenesis assay was conducted according to our previous description [42]. In brief, hFOB 1.19 cells were incubated with osteogenic inducing medium containing 10 nM dexamethasone, 10 mM β-glycerophosphate, and 50 µg/mL ascorbic acid in the presence of various concentrations of PL for 10 days. The cells were fixed with 4% paraformaldehyde and then stained for ALP activity.

### 4.6. Bone Resorption Assay

Fresh bovine femur was purchased from a local butcher. The muscles were removed and the femur was washed with PBS. The cortical bone was cut into 500-µm-thick slices using a Buhler Isomet low-speed saw (Buehler Ltd., Lake Bluff, IL, USA). The bovine bone slices were sterilized with 70% ethanol and then rinsed with sterile PBS. The sterile and dried bone slices were put into 96-well plates and soaked into α-MEM medium at 37 °C overnight. BMM cells (8 × 10^3^ cells/well) were seeded on these bone slices and treated with M-CSF (30 ng/mL) and RANKL (100 ng/mL) for 3~4 days until osteoclasts begin to form. Cells were then dissociated and seeded onto the bone slices at a density of 3 × 10^4^ per well. These cells were continuously incubated with RANKL and different concentrations of PL (0, 0.25, 0.5, and 1 µM) for another 48 h. After gently brushing the osteoclasts, the bone resorption was visualized by a scanning electron microscopy (JSM 5610 LV; JEOL, Tokyo, Japan). Six view fields were randomly selected for each bone slice for further analysis. The areas of bone resorbed were determined using Image Pro-Plus 5.0 software (Media Cybernetics, Silver Spring, MD, USA). The result was shown as resorption areas relative to the total areas of bone.

### 4.7. F-Actin Ring-Formation Assay

F-actin ring-formation assay was performed according to our previous description [4]. Mature osteoclasts cultured on the bone slices were fixed with 4% paraformaldehyde for 15 min and then permeabilized with 0.5% Triton X-100 for 10 min. F-actin rings were stained with rhodamine-conjugated phalloidin and the nuclei were dyed with DAPI solution. Fluorescence images were acquired using a Nikon resonance scanning confocal microscope (Nikon, Tokyo, Japan). The numbers of F-actin rings and actin ring surface/cell surface were analyzed using ImageJ software (NIH, Bethesda, MD, USA).

### 4.8. Western Blot Analysis

To detect the effect of PL on MAPK and NF-κB signaling pathways, BMM cells (4 × 10^4^ cells/well) were plated into 12-well plates and treated with or without PL (1 μM) for 4 h. These cells were then stimulated with RANKL (100 ng/mL) for 0, 5, 10, 20, 30, or 60 min. To detect the effect of PL on the protein expression of NFATc1, BMM cells were incubated with PL (1 µM), RANKL (100 ng/mL), and M-CSF (30 ng/mL) for 0, 1, 3, or 5 days. The cells were washed with PBS, and lysed with RIPA lysis buffer (Beyotime Institute of Biotechnology, Shanghai, China) containing protease and phosphatase inhibitor cocktail (Beyotime Institute of Biotechnology). The protein was collected by centrifugation and separated by sodium dodecyl sulfate-polyacrylamide gel electrophoresis. Subsequently, the protein was transferred to polyvinylidene difluoride membranes (Bio-Rad, Hercules, CA, USA) and the membranes were incubated with relevant primary antibodies at 4 °C overnight. After washing three times with TBST, the membranes were incubated with corresponding secondary antibodies for 50 min at room temperature. The protein bands were detected using enhanced chemiluminescence reagents (Amersham, Shanghai, China), and the relative expression of each protein was analyzed using ImageJ software.

### 4.9. RNA Extraction and Quantitative Real-Time Polymerase Chain Reaction (X)

BMM cells (4 × 10^4^ cells/well) were seeded into 12-well plates and treated with different concentrations of PL in the presence of RANKL and M-CSF for 5 days. These cells were lysed and total RNA was extracted using TRIzol reagents. Complementary DNA (cDNA) was synthesized from 500 ng of total RNA and Q-PCR was carried out using a SYBR Green Master (Vazyme Biotech Co., Ltd., Nanjing, China). The following primer sets were used: mouse cathepsin K (*CTSK*), 5′-CTTCCAATACGTGCAGCAGA-3′ (forward), 5′-TCTTCAGGGCTTTCTCGTTC-3′ (reverse); mouse *TR**AP*, 5′-CTGGAGTGCACGATGCCAGCGACA-3′ (forward), 5′-TCCGTGCTCGGCGATGGACCAGA-3′ (reverse); mouse calcitonin receptor (*CTR*), 5′-TCAGGAACCACGGAATCCTC-3′ (forward), 5′-ACATTCAAGCGGATGCGTCT-3′ (reverse); mouse *c-Fos*, 5′-CCAGTCAAGAGCATCAGCAA-3′ (forward), 5′-AAGTAGTGCAGCCCGGAGTA-3′ (reverse); mouse *NFATc1*, 5′-CCGTTGCTTCCAGAAAATAACA-3′ (forward), 5′-TGTGGGATGTGAACTCGGAA-3′ (reverse); mouse *GAPDH*, 5′-ACCACAGTCCAAGCCATCAC-3′ (forward), 5′-CACATTGGGGGTAGGAACAC-3′ (reverse). Three independent experiments were conducted with three parallel reactions.

### 4.10. Ti Particle-Induced Calvarial Osteolysis Mouse Model

Twenty C57BL/6 male mice (6–8 weeks old) were obtained from Shanghai SLAC Laboratory Animal Co., Ltd. (Shanghai, China). Mice were housed under specific pathogen-free conditions (22 °C, 50–55% humidity, 12/12 h light/dark) with free access to water and food. The animal experiments were approval by the Experimental Committee of Nanjing Normal University (^#^20201203, approved date: 3 December 2020). A Ti particle-induced calvarial osteolysis model was established per our previous description [4]. Briefly, the cranial periosteum was separated from the calvarium and Ti particles (30 mg) were embedded under the periosteum at the middle suture of the calvaria. All the animals were randomly divided into four groups: sham group (0.9% saline), vehicle-treated group (Ti particles with 0.9% saline), and PL-treated groups (Ti particles with 1.5 mg/kg/d or 3 mg/kg/d PL, respectively). One day after surgery, PL or 0.9% saline was intraperitoneally administrated once a day for 14 days. When the treatment was ended, the mice were sacrificed, and the calvarias were carefully dissected. After removing the titanium particles, these calvarias were collected for micro-CT scanning and histological analysis.

### 4.11. Micro-CT Scanning

After 4% paraformaldehyde fixation for 48 h, the calvarias were analyzed using micro-CT (SkyScan1176; Bruker, Germany) with an isometric resolution of 9 μm. After reconstruction, a circle with a diameter of 4 mm around the midline suture was selected as the region of interest (ROI) to further quantitatively analyze. Bone mineral density (BMD) and bone volume/tissue volume (BV/TV) were determined using CTan software (SkyScan, Kontich, Belgium).

### 4.12. Histological Staining and Histomorphometric Analysis

After micro-CT scanning, the calvaria samples were decalcified with 10% EDTA for a week, embedded in paraffin and stained with hematoxylin and eosin (H&E) and TRAP. The histomorphometric parameters, including the percentage of infiltrated fibrotic area against total tissue area (erosion area, %), the number of TRAP^+^ osteoclasts normalized to bone area, and the percentage of osteoclast surface per bone surface (OcS/BS, %), were calculated using Image Pro-Plus 5.0 software.

### 4.13. Statistical Analysis

Results were expressed as the mean ± SD with three or more independent experiments. Statistical difference was determined using one-way analysis of variance (ANOVA), followed by Tukey post hoc analysis. *p* < 0.05 was designated as statistically significant.

## Figures and Tables

**Figure 1 ijms-23-02868-f001:**
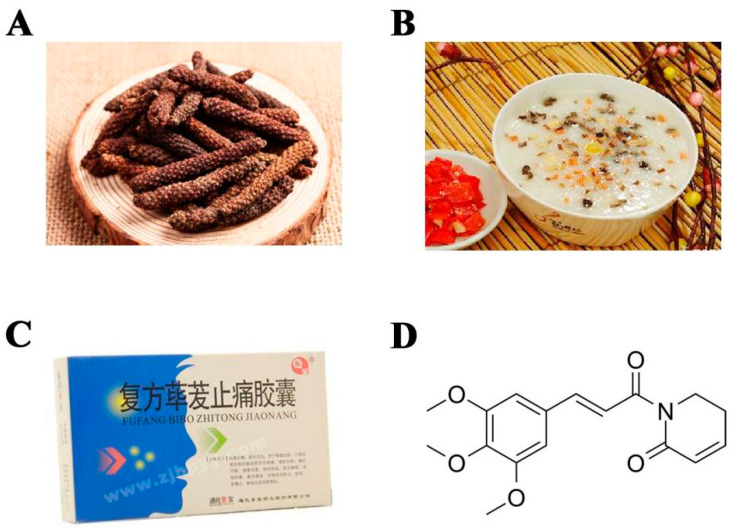
*Piper longum* L. and piperlongumine (PL). (**A**) The fruits of *Piper longum* L. (**B**) The fruits can be used to cook congee as a seasoning ingredient. (**C**) The medicinal value of *Piper longum* L. FUFANG BIBO ZHITONG JIAONANG: compound analgesic capsules containing *Piper longum* L. (**D**) The molecular structure of PL.

**Figure 2 ijms-23-02868-f002:**
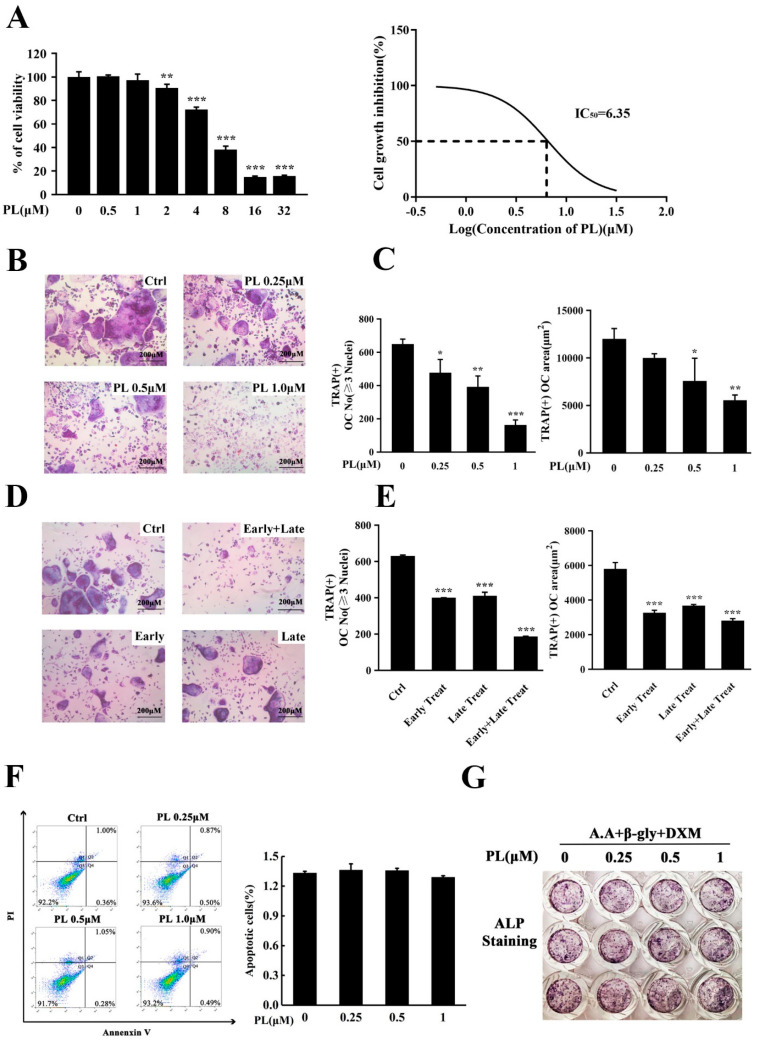
PL inhibited RANKL-induced osteoclastogenesis. (**A**) An MTS assay was performed to measure the cytotoxic effect of PL on BMM cells. The IC_50_ of PL for BMM cells was calculated. *n* = 3, ** *p* < 0.01 and *** *p* < 0.001 relative to PL-untreated control. (**B**) BMM cells were seeded into 96-well plates and treated with various doses of PL (0, 0.25, 0.5, and 1 µM) in the presence of macrophage colony-stimulating factor (M-CSF, 30 ng/mL) and RANKL (100 ng/mL) for five days. After fixing with 4% paraformaldehyde, the cells were stained for TRAP activity. (**C**) The number and area of TRAP^+^ cells (≥3 nuclei) were counted. *n* = 3, * *p* < 0.05, ** *p* < 0.01 and *** *p* < 0.001 relative to RANKL-induced, PL-untreated control. (**D**,**E**) PL (1 µM) was added at different differentiation time points (Ctrl: RANKL stimulation for five days without PL; Early Treat: co-incubation with RANKL and PL for two days and RANKL continuous incubation for another three days; Late Treat: RANKL incubation for two days and subsequent co-incubation with RANKL and PL for three days; Early+Late Treat: co-incubation with RANKL and PL for five days). TRAP staining was carried out to quantify the number of TRAP^+^ osteoclasts (≥3 nuclei). *n* = 3, *** *p* < 0.001 relative to control. (**F**) Flow cytometry was used to detect the effect of PL on apoptosis. (**G**) Human osteoblasts (hFOB 1.19) were treated with different doses of PL in the presence of ascorbic acid (A.A), β-glycerophosphate (β-gly), and dexamethasone (DXM) for 10 days. The cells were fixed with 4% paraformaldehyde and then stained for alkaline phosphatase (ALP) activity.

**Figure 3 ijms-23-02868-f003:**
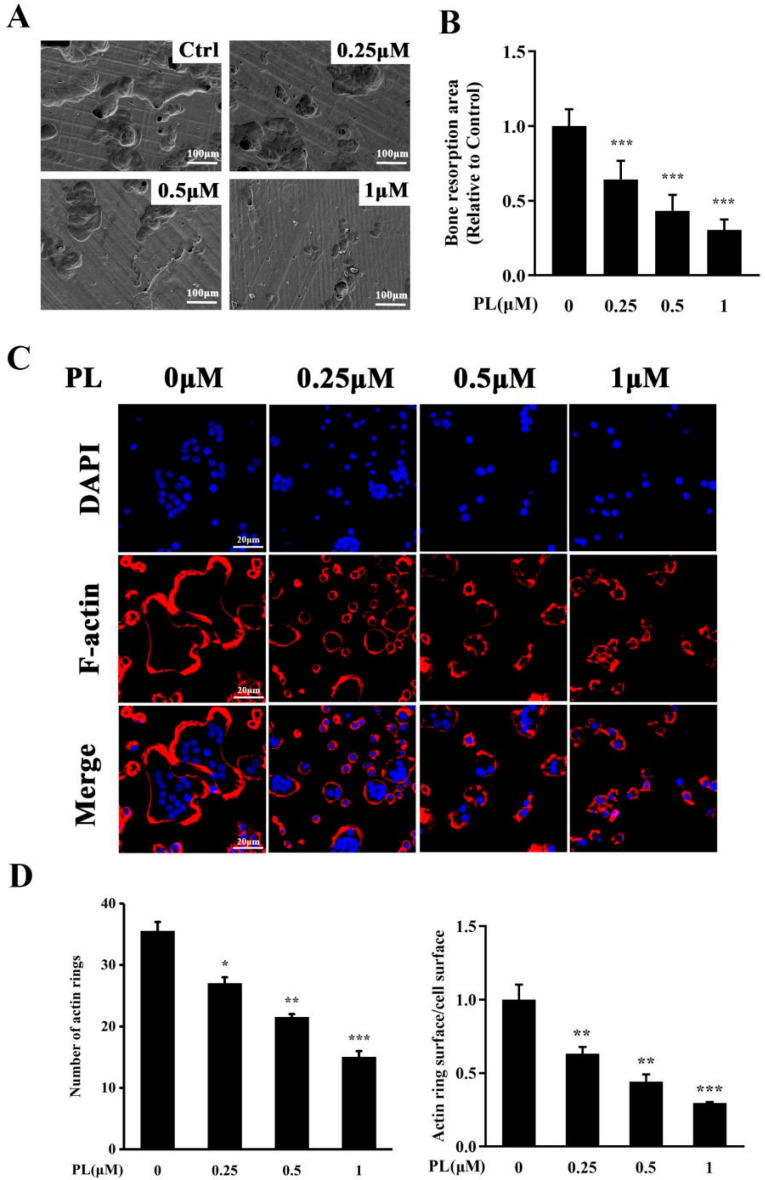
PL suppressed the formation of F-actin ring and osteoclastic resorption function. BMMs were stimulated with M-CSF and RANKL for three–four days until osteoclasts begin to form. These osteoclasts were dissociated, seeded onto bone slices, and incubated with RANKL for another 48 h. (**A**) After removal of osteoclasts, bone resorption was visualized by scanning electron microscopy. (**B**) The resorption areas were calculated using ImageJ software. *n* = 3, *** *p* < 0.001 relative to RANKL-induced, PL-untreated group. (**C**) The effect of PL on F-actin ring formation was assessed using immunofluorescent staining. (**D**) The numbers of F-actin rings and actin ring surface/cell surface were calculated. *n* = 3, * *p* < 0.05, ** *p* < 0.01 and *** *p* < 0.001 relative to RANKL-induced, PL-untreated group.

**Figure 4 ijms-23-02868-f004:**
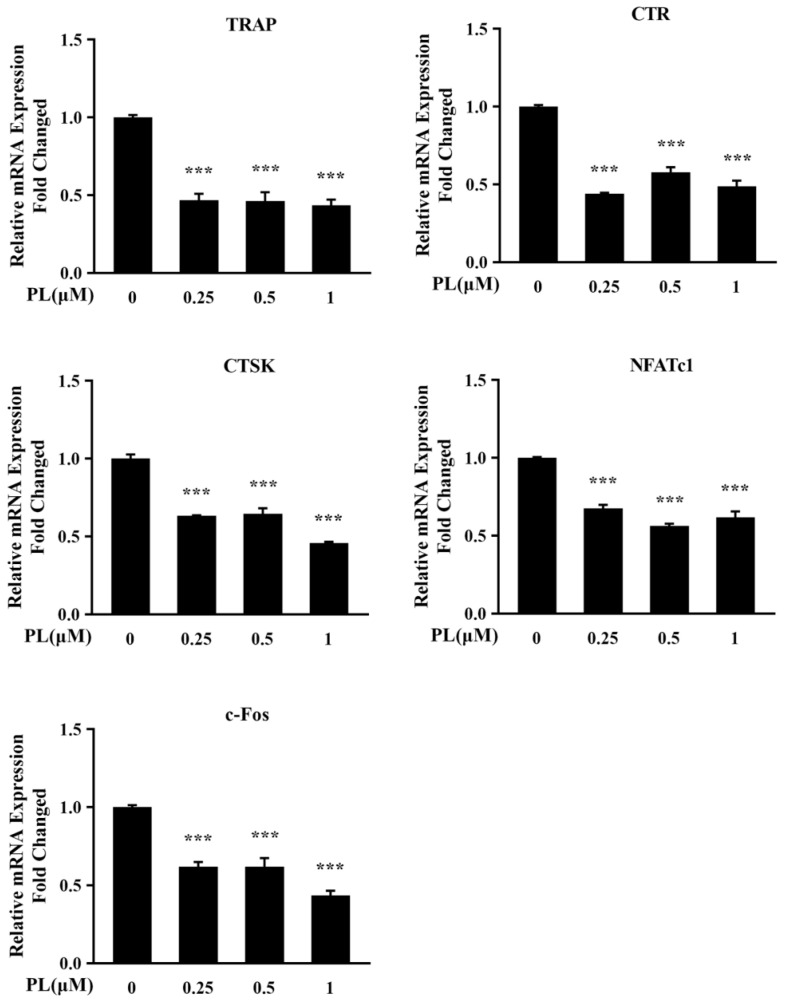
PL decreased osteoclast-associated genes expression. BMM cells were treated with different doses of PL containing RANKL (100 ng/mL) and M-CSF (30 ng/mL) for five days. Q-PCR was performed to examine the transcripts of osteoclastic marker genes. The mRNA levels of these genes were normalized to *GAPDH*. *n* = 3, *** *p* < 0.001 relative to RANKL-induced, PL-untreated group. TRAP, tartrateresistant acid phosphatase; CTR, calcitonin receptor; CTSK, cathepsin K; NFATc1, nuclear factor of activated T-cells cytoplasmic 1; GAPDH, glyceraldehyde-3-phosphate dehydrogenase.

**Figure 5 ijms-23-02868-f005:**
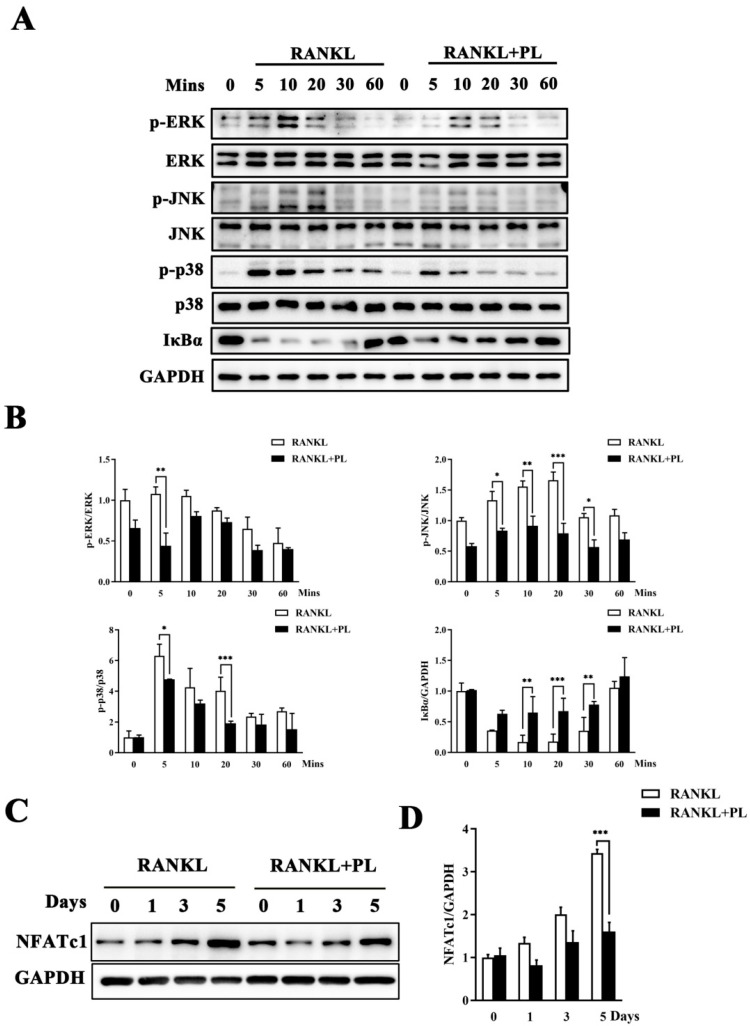
PL suppressed RANKL-induced activations of MAPK and NF-κB signaling pathways. (**A**) BMM cells were pre-treated with or without PL (1 μM) for four hours, and then stimulated with RANKL (100 ng/mL) for 0, 5, 10, 20, 30, or 60 min. Protein was extracted for Western blot with the indicated antibodies. (**B**) The ratios of p-JNK/JNK, p-ERK/ERK, p-p38/p38, and IκBα/GAPDH were determined using ImageJ software. *n* = 3, * *p* < 0.05, ** *p* < 0.01, and *** *p* < 0.001. (**C**) BMM cells were treated with PL (1 µM) for 0, 1, 3, or 5 days. Western blot was used to examine the protein levels of NFATc1 and GAPDH. (**D**) The ratio of NFATc1/GAPDH was analyzed using ImageJ software. *n* = 3, *** *p* < 0.001.

**Figure 6 ijms-23-02868-f006:**
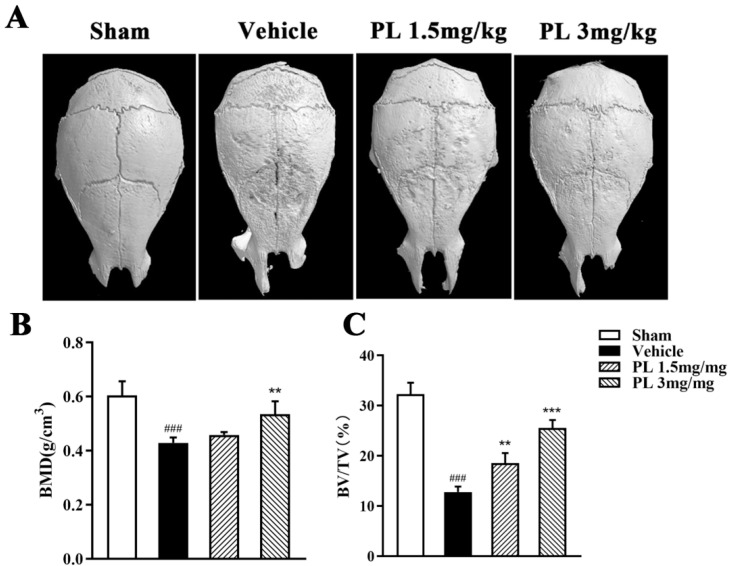
PL protected against titanium (Ti) particle-induced calvarial osteolysis. (**A**) Representative 3D reconstruction of the whole calvarias from different groups. (**B**,**C**) The quantitative analysis of bone mineral density (BMD) and bone volume/tissue volume (BV/TV). *n* = 5, ### *p* < 0.001 relative to the sham group; ** *p* < 0.01 and *** *p* < 0.001 relative to the vehicle group. (**D**) Representative images of calvarial sections stained with H&E and TRAP from different groups. (**E**,**F**) Histomorphometric analysis of the percentage of infiltrated fibrotic area against total tissue area (erosion area, %), and the percentage of osteoclast surface per bone surface (OcS/BS, %). *n* = 5, ### *p* < 0.001 relative to the sham group; *** *p* < 0.001 relative to the vehicle group.

## Data Availability

The raw data supporting the conclusion of this article will be made available by the authors, without undue reservation, to any qualified researcher.

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
