# Peer review of "Piperlongumine Inhibits Titanium Particles-Induced Osteolysis, Osteoclast Formation, and RANKL-Induced Signaling Pathways"

_ijms, 2022, doi:10.3390/ijms23052868_

Round 1

Reviewer 1 Report

Authors analyzed the effects of PL on osteoclasts affectin RANKL pathways.

I suggest to add data on osteoblasts due to their production of RANKL. In vivo the crosstalk is fundamental for nomral bone remodeling. 

About the actin ring, I cannot appreciate the reduction after treatment. It seems like the actin ring surface is the same but the cell dimension is reduced. So, Adjust dtaa on cell dimension (actin ring surface/cell surface).

Add micron scale on resorption analysis. You can notice a difference in terms of PIT vs TRENCH?

Moreover, specify the MNCs int he Y axis, what stand for?

Author Response

Point 1: I suggest to add data on osteoblasts due to their production of RANKL. In vivo the crosstalk is fundamental for nomral bone remodeling.

Response 1: We thank the reviewer for this important suggestion. In the revision work, we have assessed the effect of PL on osteoblast formation using in vitro osteoblastogenesis assay. The result showed that PL had little effect on osteoblast differentiation. The related data have been shown in Fig. 2G. And the corresponding descriptions in "Results", "Figure legends" and "Methods" sections were also presented in Page 3, Line 84-90; Page 5, Line 108-111 and Page 13, Line 314-322, respectively.

Point 2: About the actin ring, I cannot appreciate the reduction after treatment. It seems like the actin ring surface is the same but the cell dimension is reduced. So, Adjust dtaa on cell dimension (actin ring surface/cell surface).

Response 2: In the revision work, we have re-analyzed the sizes of F-actin rings. The data have been adjusted as actin ring surface/cell surface. Please see Fig. 3D. The corresponding descriptions in "Results", "Figure legends" and "Methods" sections were presented in Page 5, Line 123-124; Page 6, Line 134 and Page 13, Line 345-346, respectively.

Point 3: Add micron scale on resorption analysis. You can notice a difference in terms of PIT vs TRENCH?

Response 3: We thank the reviewer for this suggest. Indeed, both resorption trenches and resorption pits could be clearly observed on the bone slices. In the control group, the extent of trenches reached approximately 60% of the resorbed surface. PL treatment significantly decreased the formation of bone resorption, especially the formation of trenches. The related descriptions have been added in Page 5 Line 115-119.  

Point 4: Moreover, specify the MNCs in the Y axis, what stand for?

Response 4: MNCs stand for TRAP(+) osteoclasts (≥3 nuclei). The title of Y axis has been changed in the revision work. Please see Figure 2C and 2E.

Reviewer 2 Report

In this study, Liu et al. examined the effect of piperlongumine (PL), an alkaloid compound from Piper longum L fruits on in vitro osteoclastogenesis and in vivo titanium particle-induced osteolysis. The authors showed that PL treatment dose dependently inhibited osteoclastogenesis in terms of number and area of multinucleated TRAP+ cells. PL treatment inhibited osteoclastic bone resorption and actin-ring formation. PL treatment decreased expression levels of osteoclastic genes such as TRAP, CTR, and CTK and early osteoclast differentiation factors such as c-Fos and NFATc1. To understand molecular mechanisms underlying the effects of PL, the authors examined RANKL-dependent signaling pathways including ERK, JNK, p38, and NF-kB, and showed impaired induction of these signaling pathways. Additionally, the authors showed that PL treatment attenuated RANKL-induced protein expression of NFATc1. Finally, the authors applied PL to treat titanium particle-induced osteolysis. Systemic injection of PL significantly inhibited titanium particle-induced osteolysis, along with reduced osteoclastic bone resorption. Taken together, the authors concluded that PL has the potential for treatment of osteoclast-related osteolytic diseases. I have several questions and comments about this study.

  1. PL treatment (1 uM) in early phase (2 days) and/or late phase (last 3 days) showed same inhibitory effect on osteoclast differentiation, although the inhibitory effect was less prominent than whole term (5 days) treatment (Figure 2). Given that the molecular events invoked during early phase and late phase are different, the molecular effect of PL treatment may be varied. To gain insights about this issue, osteoclastic gene expression profiles (Fig 4) should be shown as day-by-day (from Day 0 to Day 5) but not as PL dose dependency. Also, RANKL-induced signaling pathways (Figure 5) need to be addressed using not only BMMs but also preosteoclasts (BMMs 2-3 days treated with RANKL. They are committed to be mature osteoclasts, yet mononucleated or less multinucleated).
  2. In Figure 2B and C, PL treatment inhibited osteoclast differentiation in a dose dependent manner, and it looked same trend in Figure 3D. On the other hand, Figure 3A showed no inhibitory effect of PL on osteoclast differentiation. These data are conflicted and do not make sense. To asses bone resorption activities per osteoclasts, the authors have to first induce mature osteoclasts, harvest them, split onto bone slices evenly in the presence or absence of PL, culture appropriate period, then remove all cells and measure resorption pits.
  3. It has been reported that titanium-particle regulates osteoblast differentiation and bone formation (DOI: 10.1038/cddis.2017.275, DOI: 10.1016/j.biomaterials.2012.03.005). It should be better to assess the effect of PL on osteoblast differentiation and bone formation to interpret PL effects on titanium particle-induced osteolysis precisely.

Author Response

Point 1: PL treatment (1 uM) in early phase (2 days) and/or late phase (last 3 days) showed same inhibitory effect on osteoclast differentiation, although the inhibitory effect was less prominent than whole term (5 days) treatment (Figure 2). Given that the molecular events invoked during early phase and late phase are different, the molecular effect of PL treatment may be varied. To gain insights about this issue, osteoclastic gene expression profiles (Fig 4) should be shown as day-by-day (from Day 0 to Day 5) but not as PL dose dependency. Also, RANKL-induced signaling pathways (Figure 5) need to be addressed using not only BMMs but also preosteoclasts (BMMs 2-3 days treated with RANKL. They are committed to be mature osteoclasts, yet mononucleated or less multinucleated).

Response 1: Indeed, the molecular events invoked during early phase of osteoclastogenesis and late phase are different. In the early stage, RANK signaling is mediated by recruiting adaptor molecule such as TRAF6, which leads to the activation of MAPKs, NF-κB and AP-1. Activated MAPKs and NF-κB induce NFATc1, which is the key osteoclastogenesis regulator. In the late stage of osteoclastogenesis, the co-stimulatory signal induces Ca2+ oscillation via PLCγ2 together with c-Fos/AP-1, wherein Ca2+ signaling facilitates the robust production of NFATc1. Subsequently, NFATc1 translocates into the nucleus where it induces numerous osteoclast-specific target genes that are responsible for cell fusion and function. In this present study, although early phage and late phage of osteoclastogenesis were both inhibited by the compound, we laid emphasis on revealing its molecular mechanism of action on early differentiation but not late differentiation. This experimental scheme is in consistence with that of many published papers. We have provided a few references to support our experimental designs.

Hu, B., Wu, F., Shi, Z., He, B., Zhao, X., Wu, H., & Yan, S. (2019). Dehydrocostus lactone attenuates osteoclastogenesis and osteoclast-induced bone loss by modulating NF-κB signalling pathway. J Cell Mol Med, 23(8), 5762-5770. doi:10.1111/jcmm.14492

Kwak, S. C., Cheon, Y. H., Lee, C. H., Jun, H. Y., Yoon, K. H., Lee, M. S., & Kim, J. Y. (2020). Grape Seed Proanthocyanidin Extract Prevents Bone Loss via Regulation of Osteoclast Differentiation, Apoptosis, and Proliferation. Nutrients, 12(10). doi:10.3390/nu12103164

Li, Y., Lin, S., Liu, P., Huang, J., Qiu, J., Wen, Z., Zhang, S. (2021). Carnosol suppresses RANKL-induced osteoclastogenesis and attenuates titanium particles-induced osteolysis. J Cell Physiol, 236(3), 1950-1966. doi:10.1002/jcp.29978

Wu, X., Zhao, K., Fang, X., Lu, F., Zhang, W., Song, X., Chen, H. (2021). Inhibition of Lipopolysaccharide-Induced Inflammatory Bone Loss by Saikosaponin D is Associated with Regulation of the RANKL/RANK Pathway. Drug Des Devel Ther, 15, 4741-4757. doi:10.2147/dddt.S334421

Although we don’t use RANKL-induced pre-osteoclasts to investigate the molecular mechanisms invoked during late phase of osteoclastogenesis, we did examine the effect of PL on induction of NFATc1. As we mentioned above, NFATc1 may represent a master switch for regulating late differentiation of osteoclasts. Additionally, we also detected the transcripts of NFATc1’s downstream target genes such as TRAP, CTR and CTSK, all of which are late differentiation marker genes. As expected, PL effectively suppressed the induction of NFATc1 and decreased the transcripts of osteoclastic marker genes. Certainly, in these above-mentioned molecular mechanisms, we don’t exclude the molecular effect of PL on early phase of osteoclastogenesis. Further investigations are needed to unveil the molecular effect of PL on late differentiation of osteoclasts.

Point 2: In Figure 2B and C, PL treatment inhibited osteoclast differentiation in a dose dependent manner, and it looked same trend in Figure 3D. On the other hand, Figure 3A showed no inhibitory effect of PL on osteoclast differentiation. These data are conflicted and do not make sense. To asses bone resorption activities per osteoclasts, the authors have to first induce mature osteoclasts, harvest them, split onto bone slices evenly in the presence or absence of PL, culture appropriate period, then remove all cells and measure resorption pits.

Response 2: We thank the reviewer for this important suggestion. In the revision work, we re-performed the bone resorption experiment. We firstly induced BMM cells to form mature osteoclasts in collagen coated plates. We then dissociated and seeded the same number of mature osteoclasts on the bone slices. After 48 h induction, we removed all the cells and measured resorption pits using scanning electron microscopy. The related data were presented in Figure 3A and 3B. Accordingly, the descriptions in "Results", "Figure legends" and "Methods" sections were presented in Page 5, Line 115-119; Page 6, Line 128-132 and Page 13, Line 329-333, respectively. In addition, Figure 3A in the original manuscript has been deleted and all the panels of Figure 3 have been re-arranged.

Point 3: It has been reported that titanium-particle regulates osteoblast differentiation and bone formation (DOI: 10.1038/cddis.2017.275, DOI: 10.1016/j.biomaterials.2012.03.005). It should be better to assess the effect of PL on osteoblast differentiation and bone formation to interpret PL effects on titanium particle-induced osteolysis precisely.

Response 3: We thank the reviewer for this important suggestion. In the revision work, we have assessed the effect of PL on osteoblast formation using in vitro osteoblastogenesis assay. The result showed that PL had little effect on osteoblast differentiation. Thus, in this present study, we focused on osteoclast differentiation to interpret PL’s effect on Ti particle-induced osteolysis. The osteoblastogenesis data have been presented in Fig. 2G. And the corresponding descriptions in "Results", "Figure legends" and "Methods" sections were presented in Page 3, Line 84-90; Page 5, Line 108-111 and Page 13, Line 314-322, respectively.

Round 2

Reviewer 1 Report

Thanks, you addressed all my suggestions.

Author Response

Response to Reviewer 1 Comments:

 We really appreciate for your positive approval on our manuscript. Thanks a lot.

Reviewer 2 Report

The authors performed experiments to answer my questions.

I suggest the authors to add discussion describing about point 1 in the main text (maybe in Discussion). In others, I'm satisfied.

Author Response

Response to Reviewer 2 Comments:

Thank you for your valuable suggestions on the manuscript. We have now added some descriptions in Discussion section. Please see Page 11, Line 232-244; Page 12, Line 263-269 and Line 270-272.